# Phototransduction in Anuran Green Rods: Origins of Extra-Sensitivity

**DOI:** 10.3390/ijms222413400

**Published:** 2021-12-13

**Authors:** Luba A. Astakhova, Artem D. Novoselov, Maria E. Ermolaeva, Michael L. Firsov, Alexander Yu. Rotov

**Affiliations:** Sechenov Institute of Evolutionary Physiology and Biochemistry, 44 Torez Prospect, 194223 St. Petersburg, Russia; lubkins@yandex.ru (L.A.A.); novosiolov.art@mail.ru (A.D.N.); me_ermlva@mail.ru (M.E.E.); michael.firsov@gmail.com (M.L.F.)

**Keywords:** photoreceptor, green rod, anuran, phototransduction, light sensitivity, dark noise, nocturnal vision

## Abstract

Green rods (GRs) represent a unique type of photoreceptor to be found in the retinas of anuran amphibians. These cells harbor a cone-specific blue-sensitive visual pigment but exhibit morphology of the outer segment typical for classic red rods (RRs), which makes them a perspective model object for studying cone–rod transmutation. In the present study, we performed detailed electrophysiological examination of the light sensitivity, response kinetics and parameters of discrete and continuous dark noise in GRs of the two anuran species: cane toad and marsh frog. Our results confirm that anuran GRs are highly specialized nocturnal vision receptors. Moreover, their rate of phototransduction quenching appeared to be about two-times slower than in RRs, which makes them even more efficient single photon detectors. The operating intensity ranges for two rod types widely overlap supposedly allowing amphibians to discriminate colors in the scotopic region. Unexpectedly for typical cone pigments but in line with some previous reports, the spontaneous isomerization rate of the GR visual pigment was found to be the same as for rhodopsin of RRs. Thus, our results expand the knowledge on anuran GRs and show that these are even more specialized single photon catchers than RRs, which allows us to assign them a status of “super-rods”.

## 1. Introduction

Anuran amphibians and several urodelian species contain a unique type of photoreceptor in their retina—so-called green rods (GRs). In contrast to classical rhodopsin-containing rods (red rods, RRs), the GRs stand out with their greenish color [1,2]. The absorbance maximum of the GR visual pigment lies within the blue part of the spectrum and is about 435 nm [3,4,5]. GRs exhibit the same morphology of outer segment (OS) as the RRs: a cylindrical shape, a relatively large size, and the separation of membrane disks from plasma membrane. Still, their inner segments (ISs) have a slender elongated shape, resembling typical cones [6,7]. This morphological contradiction has led to the suggestion that GRs represent an example of “transmuted” photoreceptor that has fallen in between the ancestral cone and typical nocturnal vision receptor, like RR, during its evolution [8].

The function of a photoreceptor (diurnal or nocturnal)—conversion of the energy of photons into the electric response—is mediated by the molecular mechanisms of phototransduction, a multistep biochemical cascade, triggered by light-sensitive visual pigment. After the absorption of a photon, the photoisomerized pigment molecule activates the G-protein transducin, which then activates phosphodiesterase (PDE) decreasing the intracellular concentration of cyclic guanosine monophosphate (cGMP) and closing the ion channels of the plasma membrane (for more detail, see the reviews [9,10,11]).

The photoreceptor’s sensitivity and response kinetics are connected to its OS morphology (this sets the operating volume for biochemical reactions and the cell’s light collecting area) and to the rod or cone specificity of its phototransduction proteins [12]. The “transmuted” nature of the GRs appeared on the protein level as well—these cells’ visual pigment belongs to a family of short wavelength-sensitive cone opsins (SWS2) [13]. Inversely, it was shown that GRs express a rod-specific isoform of transducin (at least in salamander [14]). Thus, amphibian GRs contain a mixture of cone- and rod-specific protein isoforms that makes them a perspective models for studying the cone–rod transmutation (see [15,16] for reviews).

In this context, the electrophysiological experiments can provide useful information for a deeper understanding of photoreception biochemistry. Previously, such a study on toad photoreceptors provided the main functional properties of GRs, and it was revealed that they are typical receptors of nocturnal vision responding to single photons that become saturated at moderate levels of illumination [17]. That study focused mostly on the general electrophysiological features of the GRs and did not consider the biochemistry of phototransduction due to insufficient knowledge about the key players of the signaling cascade at the time.

Toad GRs also resemble typical RRs in generating discrete dark noise–spontaneous current waves in complete darkness, which are identical in shape to the responses to single photons [18,19]. It is currently believed that spontaneous visual pigment chromophore isomerization is caused by thermal fluctuations in the molecules [20,21,22], which leads to its activation and launching of a phototransduction cascade without absorbing a photon. Cone pigments generically have the frequency of spontaneous isomerization two to three orders of magnitude higher than that of rod rhodopsin [23,24,25]; however, the most short-wavelength-sensitive ones may approach the stability of rod pigments [26,27].

Recent experiments showed that the SWS2 pigment of amphibian GRs expressed in HEK293 cells is even more stable than the analogous blue-sensitive cone pigment of other vertebrates due to a stabilizing mutation [28]. Reasoning from this fact, one can expect the frequency of isomerization-like “dark” events in the GR pigment to be comparable with that of the rhodopsin in RRs. However, two independent experimental studies of this parameter on toad isolated GRs reported results that varied 400 fold [18,29]. Such variation in results on the same species’ photoreceptors could hardly be explained by polymorphisms between the populations used in different laboratories (this problem was reviewed in [30]). Considering that dark events cannot be distinguished from real photoresponses to dim flashes and that their frequency limits the threshold sensitivity of rods, the above discrepancy prevents us from the estimation of such thresholds for GRs.

The rigorous evaluation of the light sensitivity thresholds of GRs and RRs is of great importance within the framework of scotopic color vision research. Color vision requires more than one spectral type of nocturnal photoreceptors, such as red and green rods, which can cooperate at low (scotopic) illumination. The cooperation of green and red rods was confirmed for anurans by recordings from retinal ganglion cells [31] and in behavioral experiments [32].

Thereby, GRs remain an intriguing object for investigations of the possible molecular features of their phototransduction cascade comparing to classical RRs. Here, we report the results of detailed comparative study of GRs and RRs in two anuran species, cane toad (*Rhinella marina*, former *Bufo marinus*) and marsh frog (*Pelophylax ridibundus*, former *Rana ridibunda*). These animals represent Bufonidae and Ranidae families, respectively, and further in the text, they are mentioned as “toads” and “frogs”. The molecular properties of marsh frog RRs were thoroughly described in a series of works from our lab [33,34,35]; however, to the best of our knowledge, this is the first work where its GRs are characterized. Cane toad served as a reference species since its GRs are the only ones electrophysiologically described among anuran species [17,18].

## 2. Results

### 2.1. Light Capture Properties of Green and Red Rods

As we performed the electrophysiological recordings from single photoreceptors, we stimulated them with different light-emitting diodes (LED). The green LED (λ_max_ 525 nm) served as a general-purpose stimulus during our experimental protocol, and it was important to consider that it is a much more effective stimulus for red over green rods. This brings us to the necessity of expressing stimulus strength as the number of photoactivated rhodopsins (R*). These units consider the visual pigment spectrum and absorbance.

On the other hand, the volume of OS is about two-fold greater for RRs comparing to GRs [17,19]. In order to investigate the phototransduction of RRs and GRs on the molecular level, one should express the effective stimulating light intensity independently of a particular OS volume since the same biochemical machinery initiated by the single photon absorption would produce a greater response in smaller cells [36]. Thus, throughout the text, we will provide the stimulus strength as a number of photoisomerizations in a united cell volume (R*/μm^3^) per flash or per second for prolonged stimuli (for details, see Section 4.3).

In order to estimate the photoreceptor light capture, we fitted the averaged spectrum recorded with a microspectrophotometer for each rod type of either frog or toad with standard visual pigment templates, and did not observe any deviations from the previously reported absorbance maximum values [37,38]. It must be mentioned that unlike rhodopsins, the blue-sensitive visual pigments of GRs were not described appropriately with a 100% A1-based curve and needed a 2–15% proportion of A2 chromophore to fit their slightly widened main peak.

Despite no direct evidence for the presence of A2-based pigments in anuran species studied in this work (see [38]), we suggest the possible existence of GRs containing a small amount of this chromophore. The reasoning comes from the knowledge that many anurans use the A2 chromophore in the tadpole stage [39,40], and at least one Ranidae species retains it in the adult stage (American bullfrog *Lithobates catesbeianus*, see [41]). Another possible explanation is the unusual mutation in GRs’ opsin at position 47 close to the chromophore-binding site [28], which could affect the absorbance spectra.

To define the relative spectral sensitivity of the visual pigment to the stimulus, we calculated the overlap of the pigment templates with the emission spectrum of green LED and determined the integral of the product normalized on the LED emission (Figure 1). Microspectrophotometry also allowed us to estimate values for other parameters needed to analyze our electrophysiological recordings in the context of the biochemistry of the phototransduction cascade of RRs and GRs (see Table 1).

Interestingly, for both frogs and toads, the average visual pigment absorbance was lower and dichroic rate was slightly higher in GRs than in RRs; therefore, one could suggest that the visual pigment molecule packing in the disks of OSs differs in these two types of receptors. The relative sensitivity of red and green rod visual pigments to stimulating green LED differs substantially, about 34 and 36 times for frogs and toads, respectively.

### 2.2. Responses to Brief Flashes and Sensitivity Shift

We recorded a series of responses to flashes of increasing intensities from frog and toad RRs and GRs (Figure 2A,B). The shapes of the red and green rod responses were essentially similar, although the average dark current of the latter appeared to be lower. In our experiments, the ratio of mean dark currents was 2.5 for frog red/green rods and 1.2 for toads.

As the flash intensity increased, the amplitude of the response also increased gradually until reaching saturation. A further increase of the intensity resulted in prolonging the time the cell stayed in saturation and retardation of the dark current recovery. We describe the response versus intensity function for both RRs and GRs with a Hill-type equation (Figure 2C,D):
(1)$$R\left(I_{e}\right)=R_{max}\cdot \frac{I_{e}^{h}}{I_{e}^{h}+I_{0.5}^{h}}$$

Here, *R(I_e_)* is the amplitude of the response at a certain effective intensity *I_e_*, *R_max_* is the amplitude of saturated response, *h* is the Hill coefficient, and *I*_0.5_ is the effective flash intensity that elicits a half-maximum response. The mean value of half-saturating effective intensities serves further as a sensitivity parameter for different types of rods.

It appeared that frog and toad GRs have, on average, slightly greater absolute sensitivity than RRs—1.9 and 1.4 times, respectively. It should be noted that, due to a large scatter of *I*_0.5_ for toad red and green rods (SD/mean ratio > 40%), it turned out that their sensitivity did not significantly differ statistically, although the “best” of the recorded GRs had a sensitivity up to four-times higher than the average RRs. No differences in the steepness of the Hill function were observed between red and green rods of the same species, but the toad photoreceptors demonstrated generally shallower response–intensity curves compared with frog ones. Summary results on the flash response analysis are shown in Table 2.

The absolute sensitivity of both types of toad rods was approximately 1.5–2-fold higher than that of the corresponding frog photoreceptors, which depicts the high interspecies variability of this parameter. Nevertheless, the sensitivity shift between red and green rods remains, in both species, as a characteristic feature of the anuran retina. To investigate the molecular mechanisms underlying the extra-sensitivity of GRs further, we performed kinetics analysis of different phases of the photoresponse.

### 2.3. Kinetics of the Initial Photoresponse Phase and Cascade Activation

The steepness of the rising phase of photoreceptor response indicates the activation rate of the phototransduction cascade. According to the generally accepted model of biochemical amplification in phototransduction, the steepness of the initial part of response is proportional to the number of activated rhodopsins, i.e., to the stimulus intensity [9,36]. As postulated by the authors of the model, in rods at the very beginning of a current response (200–300 ms stretch), turn-off processes can be neglected, and the true amplification constant can be elicited by fitting the data points with a parabolic function.

However, it was shown that, even at such short times from the start of response, the cascade quenching processes take place, thus, distorting the value of the amplification coefficient [34]. Considering this problem, we suggested that, within the framework of this study, we could determine only the relative difference in the activation rate between RRs and GRs. Thus, we only extracted the ratio of two activation rate values by adjusting the rising phases of two responses.

We applied this analysis on responses to weak flashes with an amplitude at 15–30% of the saturation level. Since the steepness of the initial part is proportional to the flash intensity within the non-saturating range [36], we corrected the responses by adjusting them all to the same stimulus strength. As a control record, we take the average of several frog or toad RR responses after intensity adjustment (Figure 3A,B). Therefore, we adjusted the intensity-corrected response of GRs and RRs to the mean response of the RRs of the corresponding species. The adjustment coefficients give us the ratio of activation rates, and the samples for red and green rods ratios were compared. Statistical analysis showed that, for both frog and toad photoreceptors, the relative activation rates for RRs and GRs did not differ significantly between each other (Figure 3C).

We also investigated the photoresponse delay, a distinct phase before the response initial part that corresponds to the multistep biochemical process that lead to the generation of the active transducin-PDE complexes. The duration of delay phase is restricted to investigator’s ability to detect the minimum deviation of response from the pre-stimulus dark current level. Clearly, it also depends on the steepness of the initial phase and thus decreases with increasing stimulus intensity until reaching the minimum with the saturation of response rising phase [42].

Thus, we compared the minimum delay values for red and green rods’ responses to saturating flashes that excites about 0.5–1 visual pigment molecule per μm^3^. To precisely determine the delay value we averaged 40–60 responses to very bright flashes, and this procedure might lead sometimes to degradation of physiological state of the cell. Thus, we took into delay analysis only the “best” cells, which endured the recording procedure without significant decline of response amplitude.

Measured values of the minimum delay in anuran photoreceptors are presented in Table 2. They vary in range of 10 to 17 ms, which is in good agreement with our previous data on frog red rods [43]. Pairwise comparison of the delay between red and green rods showed that they does not statistically differ for both frog and toad. Thus, the kinetics of the initial stages of the photoresponse are very similar in RRs and GRs, so we can conclude, that this activation phase does not define the difference in their sensitivity.

### 2.4. Kinetics of the Photoresponse Decay Phase and Cascade Quenching

Phototransduction cascade quenching is a multistep process involving separate mechanisms for turning off the activity of visual pigment, transducin, and PDE, which are additionally regulated by calcium feedbacks [33]. To roughly estimate the rate of the reactions underlying this stage, we fitted the decay phase of non-saturated photoresponses with a single exponential function with time constant τ_off_. It appeared that the turn off in GRs is on average slower than in RRs (time constants were 1.3 s vs. 0.9 s for frogs and 2.3 s vs. 0.9 s for toads, respectively, see Figure 4). Thus, difference in sensitivity for the two types of rods is associated with the slower kinetics of turning off phase of the GRs. We also measured the time to peak and integration time for red and green rods and as expected, their values appeared longer for GRs (see Table 2).

Prolonged integration time of the response should further increase the sensitivity of GR to continuous illumination. For an accurate comparison, we recorded the series of responses to the 30-s light steps of increasing intensities, from a level of 10–20% of dark current to fully saturated one. Responses of both red and green rods to steps of moderate intensity comprised of only flat plateau phase (Figure 5A,B). In responses to higher intensities, the initial peak appears with following decrease phase that reflects the process of partial light adaptation.

In accordance with expectations, difference of sensitivity to steps of light between red and green rods was about 4–7-fold in favor of the latter (Figure 5C,D). Response vs. intensity curves for rods initial peak and plateau were well fitted by Hill function (Equation (1)). Note that for peak-approximating curves the Hill coefficients were close to 1, while curves for plateaus were much shallower (*h* = 0.5–0.7) making a widely overlapping operational range of intensities for anuran RRs and GRs.

### 2.5. Photoreceptor Discrete and Continuous Dark Noise

To reconsider previously reported data on the discrete events frequency we analyzed long-length (15–30 min) records of the dark current of red and green rods of frogs and toads. The average frequency of discrete dark waves was determined by dark current probability density histogram analysis (see Section 4.4). The histograms usually had an asymmetric shoulder and were fitted by the function
(2)Pi=A·∫01fexp(i−k*rt22σ2)dt

Here, *A* is a normalizing factor, *σ* is the dispersion of the continuous noise, *r*(*t*) is a single photon response (SPR) and *k* is the scaling factor used to correct the SPR amplitude in order to obtain an appropriate fit (see Section 3.3). Thus, *p*(*i*) is expressed as the probability density histogram of the SPR convolved with a Gaussian continuous noise (Figure 6) [21,35]. Note that corrected SPR amplitude as fraction of dark current was larger for GRs than for RRs (see Table 3), which is in a good agreement with their higher sensitivity discussed in Section 2.2.

The rate constant of spontaneous pigment activation was determined by normalizing the frequency of dark events to the number of visual pigment molecules in the cell. This number was estimated from the mean cell volume and the pigment concentrations in anuran rods calculated previously by Hárosi (3.1 mM for RRs, 3.5 mM for GRs, see [44]). Our estimate shows that in frogs and toads the GR’s visual pigment isomerization rate constant does not significantly differ from the constant of RRs’ rhodopsin (Table 3).

Despite a typical cone pigment would be expected to be about two orders of magnitude less stable than rhodopsin [28], our result is much closer to one reported by Matthews [18]. Moreover, since the volume of the outer segment of frog GRs is, on average, 1.5–1.8 times smaller than that of RRs (1275 against 2340 μm^3^ for frog, and 1520 against 2205 μm^3^ for toad) the frequency of isomerization per cell in GRs is even slightly lower compared to RRs.

It is known that the noise of rods in complete darkness has not only discrete component but is also continuous. Continuous noise consists of symmetrical low-amplitude oscillations and is likely associated with spontaneous oscillations in PDE activity [45]. We analyzed continuous noise spectra in all types of rods discussed in this work. The analysis was performed in the same way for all types of photoreceptors, described in the Section 4.4. The summarized result is shown in Figure 7. Each curve represents the averaged power spectral density of several rods (*n* = 8 for green and red frog rods, and *n* = 5 and 6 for green and red toad rods, respectively).

The spectra of all four rod types coincide within confidence limits, except for the region of 0.5–2 Hz, in which the power density spectrum of frog RRs is reliably higher than that of the other three types of rod photoreceptors. Frog RRs also shows several times higher noise levels of plasma membrane channels in the band above 2 Hz.

## 3. Discussion

### 3.1. Extraordinary Sensitivity of GRs and Its Molecular Origin

The results of our study strongly suggest that GRs of anuran amphibians are highly specialized nocturnal vision receptors. The estimated half-saturating intensities of toad GRs and RRs were somewhat lower than reported by Matthews [17]. However, Matthews reported two-fold variability in the sensitivity of RRs samples studied in 1979 and 1983 [17,19]. Thus, it seems there is no actual discrepancy between our results.

The kinetics of GR activation processes and response delay coincide with that of typical RRs. In our previous study [43], we showed that delay in phototransduction cascade is limited by the GTP-bounding transducin-PDE interaction instead of the cycle of rhodopsin-transducin interaction, which instead determines to a large extent the value of amplification gain. Since it is known that GRs express the rod-specific isoform of transducin [14] and since we found no difference in the kinetics of transducin-PDE interaction, we suggest that GRs express the same isoform of PDE as the RRs or the one with quite similar kinetic properties. This hypothesis could be verified by accurate immunohistochemical analysis of anuran PDE subunits isoforms distribution.

Phototransduction cascade quenching in GRs occurs about two-times slower than in RRs, which leads to the increase of integration time and makes them even more efficient single photon detectors. The single-exponential description of the descending phase of the photoresponse makes it possible to estimate only the summarized kinetics of the different stages of inactivation, while the particular contribution of each stage will remain out of consideration [33].

This problem can be solved by fitting the response with a detailed mathematical model containing all the main parameters of the photoresponse with their unambiguous connection with the cascade biochemistry (see [34]). Such detailed models exist for a limited number of photoreceptor types (amphibian RRs and fish cones) that are robust enough to survive during experiments with PDE inhibitors application and calcium clamp conditions needed to estimate all necessary model parameters [46,47]. However, the fragile GRs with relatively low dark current do not seem to fit this condition.

### 3.2. Cooperation of GRs and RRs in Terms of Absolute Sensitivity and under Natural Illumination

Despite the higher sensitivity of GRs compared to RRs, our results still suggest that these two photoreceptor types have a widely (up to six orders of magnitude) overlapping with the operating intensity range mostly in the scotopic region. This is in line with the recent results of behavioral experiments for common frogs (*Rana temporaria*) that preserved the ability to distinguish colors even at extremely low levels of illumination [32]. In toads, the discovered sensitivity difference between red and green rods was even smaller than in frogs, and thus the overlapping range between the two rod types should be wider. Nevertheless, the common toads (*Bufo bufo*) did not show any signs of scotopic color discrimination in those behavioral experiments and authors of the study supposed an involvement of different pathways in the analysis of color vision for each task depending on ecological relevance. Despite the numerous studies of the ability of anurans to discriminate “blue” and “yellow-green” signals at the level of ganglion cells and retinal brain projections (see [31,48,49,50]), the contribution of GRs to this process is quite questionable. All of these works were performed at photopic illumination, and the existence of anuran blue-sensitive cones (not discovered at that time [51]) spectrally indistinguishable from GRs was not considered. Thus, the neural mechanisms of anuran scotopic color vision remain mostly unknown, while here we report that there is at least a basis at the receptor level.

Comparison of the sensitivity of red and green rods to the natural illumination, like sunlight, has been done by matching the visual pigment templates with the solar spectrum. This allows us to compare the amount of photons reaching two types of anuran rods (Figure 8). Integrals of overlapping of the frog visual pigment templates with the solar spectrum give the relative spectral sensitivity for RRs–0.41 and about two times lower value for GRs S(λ)–0.18. Thereby, in the light of visual ecology, the two-fold higher absolute sensitivity of frog GRs could evolve as an adaptation to a lower proportion of short wavelength photons in the solar emission reaching the Earth.

### 3.3. Discrete Dark Noise of GRs: Reconsideration of Previously Reported Results

The frequency of discrete dark events in GRs remains an ambiguous parameter as the values previously reported by Matthews [18] and Luo et al. [29] differ by more than 400 times. Our data much better agrees with Matthews′ results on the cane toad GRs; however, our calculations gave the frequency of spontaneous isomerizations per cell about three-times lower. The latter can most likely be explained by the fact that small uncertainty in the zero position of the histogram leads to uncertainty in determining the standard deviation σ and subsequently the integral *xp(x)* (see Equations (1) and (3) in [18]), and hence an approximately two-fold variation in the resulted frequency of spontaneous events occurs. We used a slightly different approach to estimate *σ* (see Appendix B) and assume that this factor explains the small discrepancy between our and Matthews’ results fairly well. The difference between our estimates of spontaneous isomerization rate constant mostly origins from the different values of visual pigment concentration in OS used in calculations (1.4 mM in Matthews’ study, according to [52]; 3.5 mM in present work, according to [44]).

A study on toad GRs by Luo et al. [29] gave an extremely low value of the spontaneous isomerization frequency (about 150-times lower than ours), peculiar to a putative ultrastable pigment. We assume that a possible explanation for this discrepancy could be an incorrect estimate of the size of dark event. Currently, a common procedure for determining the dark event size is a Poisson analysis of a sequence of rod responses to weak light flashes [19]. This method allows us to determine the SPR and, since both SPRs and dark events arise from single visual pigment chromophore isomerization, they are supposed to have the same shape. Poisson analysis was also used in this work; however, it gave us a number of clearly overestimated values as seemed from the probability density histogram and the form of actual events from dark noise records. As a result, we could not fit the histogram with the standard function and added a scaling coefficient for the typical dark event to overcome this problem (see Equation (2) and Appendix A).

It is generally accepted that SPRs have very low variance in amplitude [53,54]. However, our group previously reported that the amplitude of discrete dark events in anuran rods varied much more than that of SPRs [35]. If so, we shall assume that variation of dark events size may has a different origin and follows different laws, than SPR does. Clearly, proving this assertion is beyond the scope of this work, but if this is the case then the size of dark events varies relatively widely, rather than being nearly fixed. It follows that the use of Poisson-determined mean amplitude of SPR may lead to underestimating of events number while counting them by the eye (as in study by Luo et al. [29]) since only the largest ones would be taken into account, and the smallest ones will be missed.

Thus, we believe that the “histogram analysis” approach used in present study gives more adequate estimation of spontaneous events frequency as it takes into account all of them with no connection of their size. We also showed that the corrected amplitude of SPR corresponding to mean discrete dark event was lower for toad GRs due to their relatively low dark current (see Table 3), while the amplitude of continuous noise in red and green rods did not differ. Hence, the eye of investigator could more easily miss large amount of small dark events masked by continuous noise in GRs.

### 3.4. High Stability of GRs’ Visual Pigment in Evolutionary Context

The properties of the visual pigment of GRs allow them to maintain a high sensitivity as spontaneous isomerization rate constant does not differ between this pigment and the rhodopsin of typical RRs. In anurans, the former belongs to family of short-wavelength sensitive cone pigments (SWS2) and, according to in vitro measurements, has acquired rod-like stability by a single amino-acid substitution [28]. Our results support this conclusion and suggest that stabilization occurred during adaptation of the GR to function as nocturnal receptor.

It has been suggested that high “noiseness” of the typical cone pigments reflects the openness of the chromophore pocket, which accelerates the exchange of chromophores during photolysis [55], and therefore one could expect the highly stable anuran GR visual pigments to decay as slowly as rhodopsins [28]. On the other hand, as inferred from the direct measurements on frog and salamander isolated GRs, the stabilized pigment retains the fast photoproduct decay kinetics typical for cone pigments [56,57,58]. Thus, it appears that native GR visual pigment combines fast decay with high stability.

The evolutionary reasons of such combination of typical rod (low isomerization rate) and cone (fast photolysis) properties apparently arises from the expression pattern of the SWS2 pigment. It was shown that tiger salamander (*Ambystoma tigrinum*) expresses this pigment in both GRs and blue-sensitive cones [14] that requires the protein to satisfy the needs of two functionally different receptor types, diurnal and nocturnal. Indeed, the spontaneous isomerization rate is lower in blue-sensitive cones compared to red-sensitive ones [26]. The low isomerization rate of the visual pigment allows GRs to effectively detect single photons while high photolysis rate allows cones to quickly remove bleached molecules from the transduction turnover and get them ready for regeneration under the bright illumination.

Taken together, our results expand the knowledge on anuran GRs, including the description of this receptor type in a new species, the marsh frog, and confirm their status as nocturnal photoreceptors. Their supposed “transmutation” from ancestral cones still remains questionable; however, our analysis showed that their specialization level for the counting of single photons was even higher than that of RRs, making them not only typical, but “super-rods”. Interestingly, the sensitivity and stability of visual pigments in tiger salamander GRs are lower than in RRs (see data from [14,26,28,59]), which may reflect the lower degree of their specialization as nocturnal receptors in this species. Therefore, the comparison of anuran and urodelian GRs can shed the light on evolutionary history of this unique photoreceptor type.

## 4. Materials and Methods

### 4.1. Experimental Animals, Preparations and Solutions

Adult marsh frogs (*P. ridibundus*) were captured in wild in southern Russia and cane toads (*R. marina*) were purchased from local breeder. Frogs were kept for up to 8 month with free access to water at 10–15 °C on a natural day–night cycle and were fed by mealworms. Toads were kept for up to 6 months with free access to water at 26–30 °C on a natural day–night cycle and fed by crickets and cockroaches.

Animals were treated in accordance with the European Communities Council Directive (24 November 1986; 86/609/EEC). Experimental protocol was approved by the local Institutional Animal Care and Use Committee. Before the experiment, animals were dark-adapted overnight. Eyes were enucleated, and the retinas were extracted under dim red light. All further procedures were conducted at dim red light or infrared TV surveillance.

The Ringer solution for frog and toad preparations and perfusion was the same and contained in mM: NaCl 90, KCl 2.5, MgCl_2_ 1.6, CaCl_2_ 1, NaHCO_3_ 5, HEPES 5, glucose 10, and EDTA 0.05; pH adjusted to 7.6. All chemicals were from Sigma-Aldrich (St. Louis, MO, USA).

### 4.2. Microspectrophotometry

To characterize spectral properties of visual pigments in the two types of rods of frogs and toads, we performed microspectrophotometric recordings from the outer segments of single photoreceptors freely floating in Ringer solution between two coverslips sealed at the edges. The design of the instrument as well as the procedures for sample preparation and recording have been described earlier [38,60]. Spectra were recorded at two polarizations of the measuring beam transversal and longitudinal with respect to the OS axis (T, transversal with respect to the OS axis, and L, longitudinal, along the axis) which allowed us to make more accurate estimation of the absorption of the nonpolarized light. To determine the maximum absorbance wavelength (λ_max_) and chromophore composition (A1 vs. A2) of visual pigments, the recorded spectra were fit with a combination of A1 and A2 templates according to [38] using least square algorithm.

### 4.3. Electrical Recordings and Light Stimulation

The responses of toad and frog rods to flashes and steps of light, as well the dark noise stretches, were registered by suction pipette recording from single cells (see [61]). Details of our experimental setup and procedures were described previously in [33,34,35]. The light stimulation system comprised of two independent channels based on high-output LEDs, the intensity in both channels was regulated under computer control stepwise by ND filters inserted in the beam and continuously by the LED current. The first channel was equipped with LED λ_max_ = 630 nm (red) and was aimed to deliver a preliminary light test to distinguish between red and green rods.

This stimulus can evoke photoresponses only in RRs while GRs stay silent even. The second channel was equipped with switchable LEDs λ_max_ = 460 and 525 nm (blue and green) and was aimed to stimulate green and red rods with short flashes of light (10 or 2 ms length) or with long steps of light (30 s length). The LEDs spectra were measured with USB4000 spectrometer (Ocean Optics, Dunedin, FL, USA). Intensities of blue and green flashes were calibrated for the majority of individual rods using the Poisson statistics of responses to weak flashes [19] and additionally with a Burr-Brown OPT-301 optosensor.

The “effective light intensity” *I_e_(λ)* expressed as number of photoisomerizations in a unite cell volume was calculated by following formula (according to [34]):*I*_*e*_(*λ*) = 2.303 * *I*(*λ*)**q***f***a*_*max*_**S*(*λ*),(3)
where *I(λ)* is the light intensity expressed as the number of photons falling on the unit square (photons/µm^2^), estimated by two independent methods (see above), and *q* = 0.67 is the quantum yield of the visual pigment bleaching [62]. Microspectrophotometry provides *a_ma_**_x_*—the peak T-density of the corresponding visual pigment (in µm^−1^), and *f* = (1 + *L*/*T*)/2, a factor correcting *a_max_* to the situation where OS is illuminated side-on with nonpolarized light, as in our electrophysiological setup (here *L/T*—dichroic rate of the visual pigment). *S(λ)* is the relative spectral sensitivity of the visual pigment to the stimulus emission.

### 4.4. Data Processing

All records were performed on single isolated rods with the outer or inner segment drawn into the pipette. The current amplitude in suction recordings depends on the quality of the seal between the pipette and the cell [61] and is usually larger in the “outer segment in” configuration than in the “inner segment in” one. Therefore, we corrected the dark current values obtained in the latter configuration by multiplying them by the coefficient 1.25 (estimated from control experiments).

Data acquisition was under the control of LabView software (National Instruments, Austin, TX, USA). Responses aimed to analyze delay were low-pass filtered at 300 Hz (8-pole analog Bessel filter) and recorded at 1-ms digitization interval. Since the delay is the time between the start of light stimulation and the deviation of response rising phase over experimental noise, we defined the latter as the first post-stimulus point that lies at or beyond the 0.0025 fractional response and is above two standard deviations of the pre-stimulus stretch. Responses aimed for other purposes were filtered at 30 Hz and recorded at 10-ms digitization.

Dark current traces were low-pass filtered at 30 Hz and recorded at 2-ms digitization. Long-lasting recordings of the dark current noise were used for calculation of frequency of spontaneous isomerization of visual pigment, and for analyzing power spectral density of continuous noise (see [21]). On this purpose, long records were cleared of slow baseline oscillations (see Appendix B, Figure A1). To perform the fitting of dark current probability density histogram the SPR was obtained from the statistics of approximately 100 responses to weak (2 to 4 R* per cell) flashes [19].

The Gaussian standard deviation σ could be determined from current traces with excised discrete dark events. Thus, the only parameters to be found by manual fitting of Equation (2) are the frequency of the SPR-like dark events, *f* (s^−1^), and the scaling coefficient for SPR, *k*. For fitting purposes, we used MathCad 15 software (PTC, Needham, MA, USA). The power spectral density of the noise was computed using the conventional Fast-Fourier transform algorithm (FFT Power Spectral Density function in LabView software). The noise spectra were calculated as the mean of spectra of 15–30 epochs.

Statistical analysis was performed in Microsoft Excel (Microsoft, Redmond, WA, USA) and GraphPad Prism 8 (GraphPad Software, San Diego, CA, USA). The normality of dataset distributions was confirmed using the Shapiro–Wilk test. Groups of red and green rod parameters were compared using the t-test with Welch’s correction, and *p* < 0.05 was taken to be statistically significant.

## Figures and Tables

**Figure 1 ijms-22-13400-f001:**
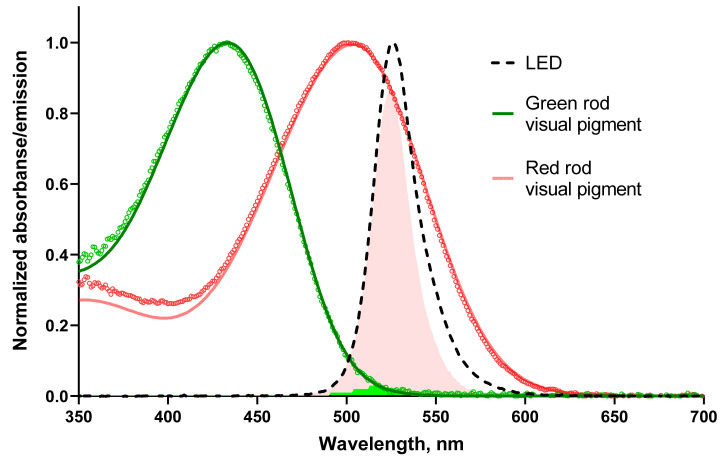
The sensitivity of frog rods to green LED used in electrophysiological experiments. The absorbance spectra of the outer segments of frog rods were recorded at polarization of the microspectrophotometer measuring beam transversal with respect to the OS axis (T-densities). Green and red open circles show unsmoothed experimental spectra averaged from 13 green rods and 12 red rods of the frog, respectively. Smooth red and green lines are the best approximations of the experimental spectra with template fits [38]. The template for red rod pigment is pure A1-based (λ_max_ 502 nm) while the pigment of green rods is fitted with a mixture of A1-95% and A2-5% (λ_max_ 433 nm). The black dashed line shows the emission spectrum of the green LED used in electrophysiological experiments. Light-red and green shadings show areas of overlapping between red or green rod visual pigment spectrum and the emission of the LED, respectively.

**Figure 2 ijms-22-13400-f002:**
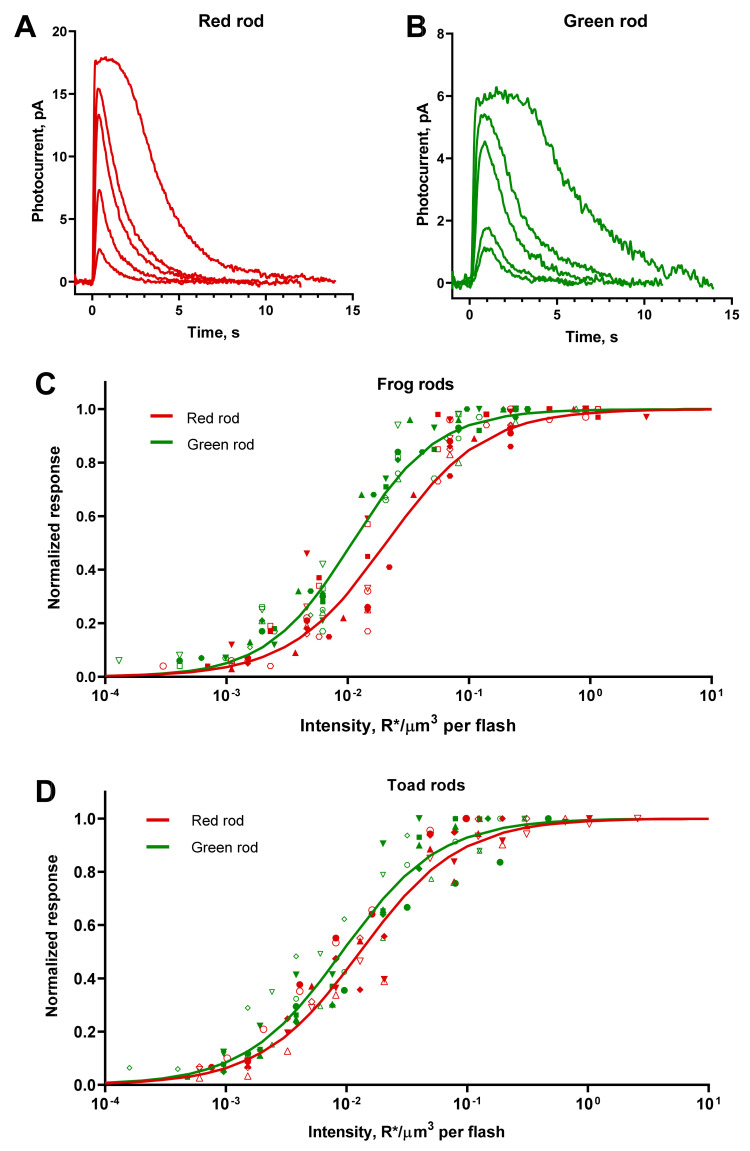
Responses of green and red rods to 10-ms flashes. (**A**,**B**) Representative current responses of frog red and green rods to 525-nm flashes of different intensities applied at moment of time 0. Flash intensities for red rods (**A**) from the weakest to strongest (R*/μm^3^ per flash): 0.007, 0.022, 0.07, 0.22, and 0.92. Flash intensities for green rods (**B**) from the weakest to strongest (R*/μm^3^ per flash): 0.002, 0.006, 0.02, 0.05, and 0.3. (**C**,**D**) Summary flash response vs. intensity curves for frog and toad rods. Each sort of symbol represents an individual cell. Smooth lines are least-square fits of all corresponding data points with Hill-type function. (**C**) Pooled data for 12 red rods and 12 green rods of frogs. The Hill fit values were *h* = 1.08, I_0.5_ = 0.018 R*/μm^3^/flash in red rods and *h* = 1.23, I_0.5_ = 0.010 R*/μm^3^/flash in green rods. (**D**) Pooled data for 8 red rods and 9 green rods of toads. The Hill fit values were *h* = 1.05, I_0.5_ = 0.011 R*/μm^3^/flash in red rods and *h* = 1.08, I_0.5_ = 0.008 R*/μm^3^/flash in green rods.

**Figure 3 ijms-22-13400-f003:**
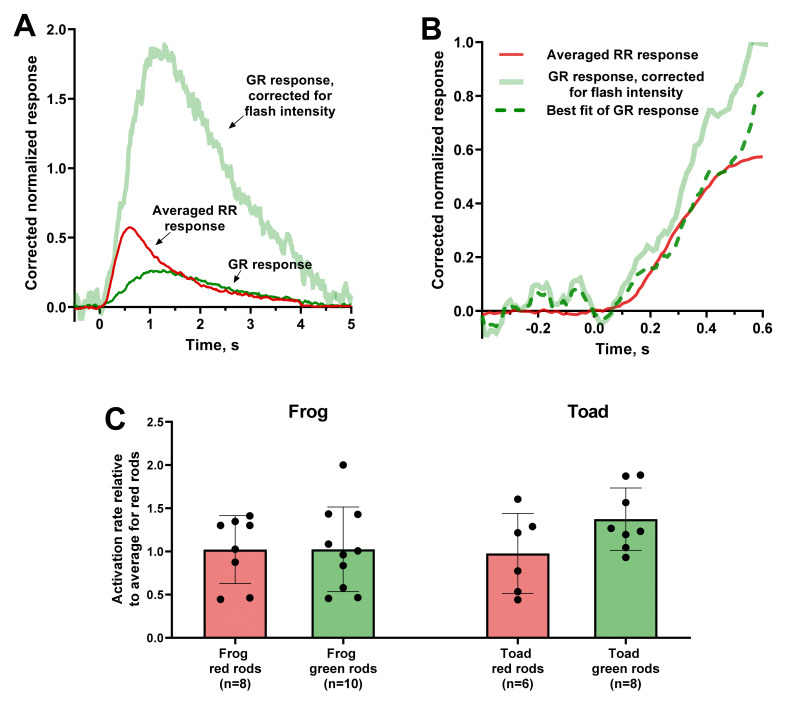
Estimation of the activation rate of red and green rods from the steepness of the fractional photoresponse rising phase. (**A**) Correction of the frog green rod response (thin green line) for the same flash intensity as the averaged response of 8 red rods (thin red line). As the intensity for red rods is 0.015 R*/μm^3^ per flash and for given green rod is 0.002 R*/μm^3^ per flash, the scaling factor for corrected response is 7.5 (thick pale green line). (**B**) Fitting of the initial phase of corrected green rod response (thick pale green line) to coincide with the averaged red rods response (red line). The dashed green line represents the best fit of corrected green rod response scaled up by a factor of 0.7. Flash was applied at moment of time 0. (**C**) Comparison of values of relative activation rates between red and green rods (reciprocals to coefficients calculated as shown on (**B**) panel). Statistically significant differences were observed neither for frog (*p* = 0.99) nor for toad (*p* = 0.11) according to a t-test with Welch’s correction. Data are presented as the means ± SD. *n*—numbers of cells in every group.

**Figure 4 ijms-22-13400-f004:**
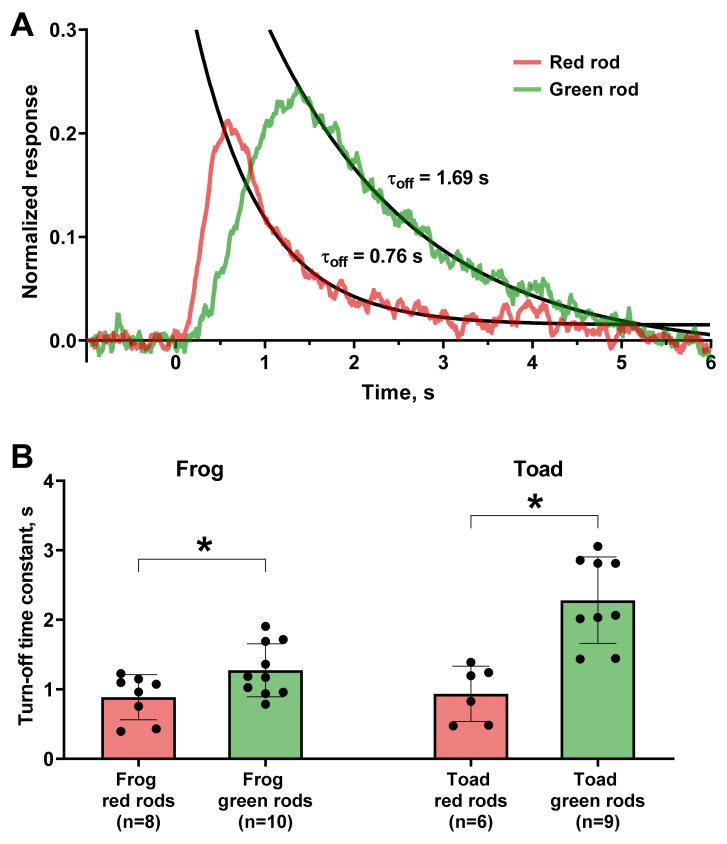
Comparison of quenching kinetics of fractional responses of red and green rods. (**A**) Representative frog red and green rods’ responses (red and green lines, respectively). Solid black lines show single exponential fit of the falling phase of two photoresponses. The turn-off time constants (τ_off_) are shown near the responses. Flash was applied at moment of time 0. (**B**) Comparison of values of turn-off time constants between red and green rods of frog and toad. *—statistically significant differences according to t-test with Welch’s correction were observed for both frog and toad cells (*p* < 0.05). Data are presented as the means ± SD. *n*—numbers of cells in every group.

**Figure 5 ijms-22-13400-f005:**
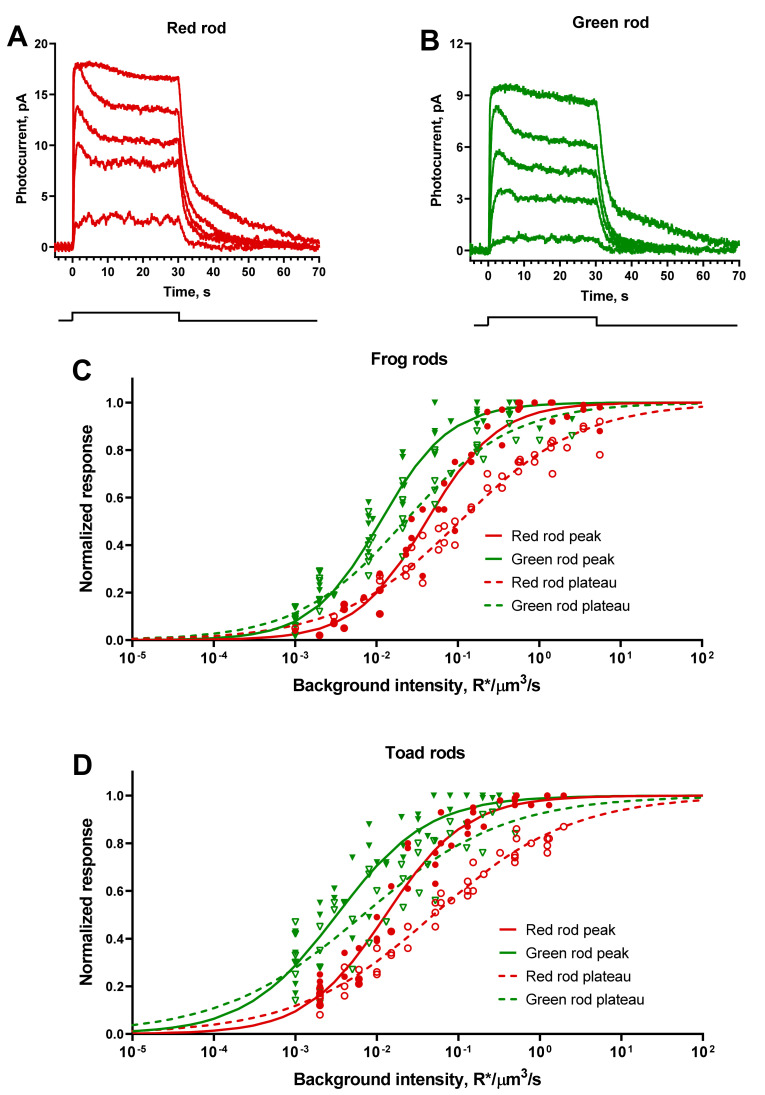
Responses of frog red and green rods to continuous light stimuli (30 s steps of light). (**A**,**B**) Representative current responses of frog red and green rods to 525-nm steps of different intensities applied at moment of time 0 (see the stimulation scheme below graphs). Step intensities for red rod (**A**), from weakest to strongest (R*/μm^3^/s): 0.007, 0.023, 0.15, 0.55, 3.49. Step intensities for green rod (**B**) from weakest to strongest (R*/μm^3^ per flash): 0.001, 0.008, 0.02, 0.05, 0.43. (**C**,**D**) Response vs. intensity functions for steps of light for frog and toad rods, pooled data for 6 red and 6 green rods for both species. Solid green and red symbols show the amplitudes of initial peaks of the responses from individual cells for green and red frog rods; open symbols show the levels of plateau just before the end of the stimulus. Solid green and red lines are least-square Hill-type fits of all corresponding data points to the peak responses, and dashed curves are fits to the plateau levels. Hill fit values: *h*  =  0.98, *I*_0.5_  =  0.041 R*/μm^3^/s in red rods and *h*  =  1.02, *I*_0.5_  =  0.011 R*/μm^3^/s in green rods (frog, peak); *h*  =  0.58, *I*_0.5_  =  0.106 in red rods and *h*  =  0.66, *I*_0.5_  =  0.022 in green rods (frog, plateau); *h * =  0.88, *I*_0.5_  =  0.013 in red rods and *h*  =  0.77, *I*_0.5_  =  0.003 in green rods (toad, peak); *h*  =  0.52, *I*_0.5_  =  0.05 in red rods and *h*  =  0.5, *I*_0.5_  =  0.007 in green rods (toad, plateau).

**Figure 6 ijms-22-13400-f006:**
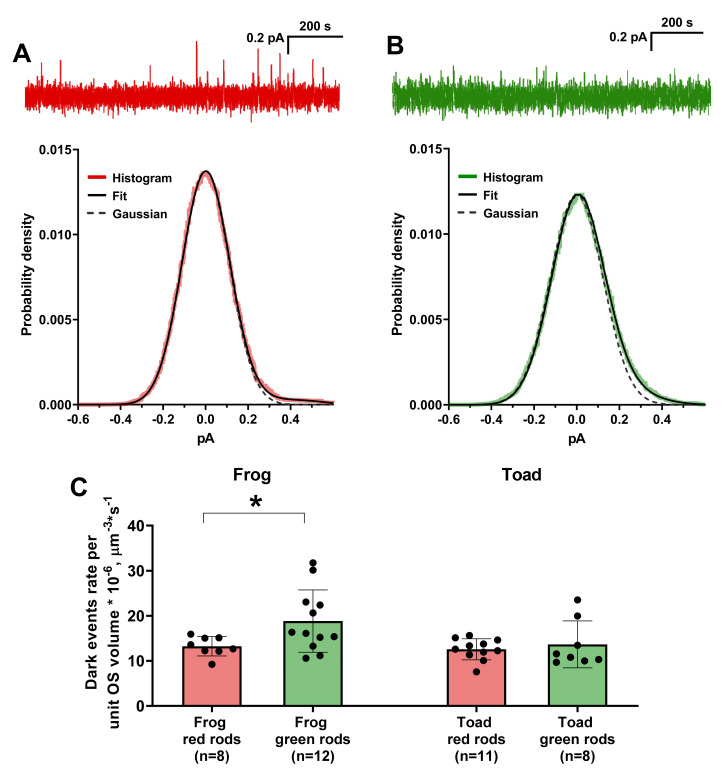
Analysis of frequency of dark events in green and red rods. (**A**,**B**) Representative stretches of dark current recordings from toad red (**A**) and green (**B**) rods and corresponding probability density functions. The red and green lines are the histograms of photoreceptor dark current probability density and dashed lines are the best fits for histograms of stretches with excised dark events by Gaussian functions (*σ* = 0.112 for red rods, *σ* = 0.123 for green rods). Black curves represent the best fits of original stretches by Gaussian functions supplemented with corresponding single photon response spectra. (**C**) Comparison of values of dark events rate per unit volume between red and green rods of frog and toad. *—statistically significant differences according to t-test with Welch’s correction were observed only for frog (*p* < 0.05). Data are presented as the means ± SD. *n*—numbers of cells in every group.

**Figure 7 ijms-22-13400-f007:**
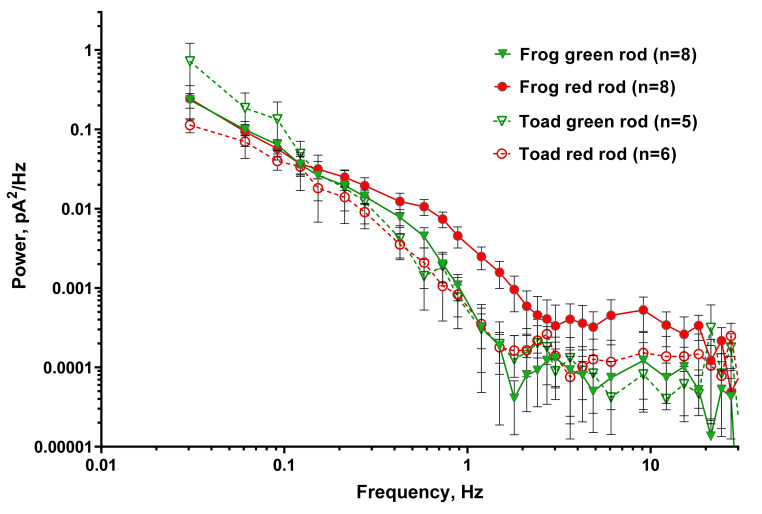
Power spectra of separated continuous dark noise for frog and toad green and red rods. Continuous noise was calculated from recordings with manually removed discrete dark events. Averaged power spectral density of several rods is shown. Solid green and red symbols show averaged power spectrum for green and red frog rods; open symbols show the power spectrum. *n*—numbers of cells in every group.

**Figure 8 ijms-22-13400-f008:**
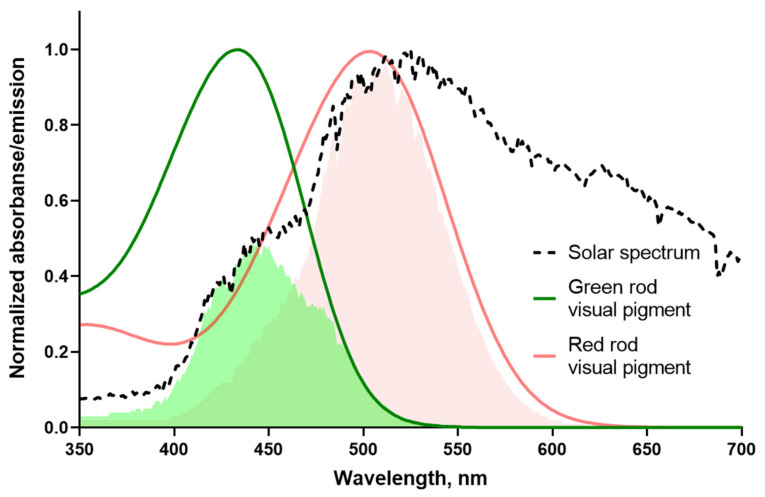
Sensitivity of frog rods to natural light. Red and green lines are template fits of experimental absorbance spectra according to [Govardovskii et al., 2000]. Template for the red rod pigment has λ_max_ = 502 nm while the pigment of green rod is fitted with λ_max_ = 433 nm. The black dashed line shows the spectrum of the white paper illuminated with direct sun, at clear sky, recorded with a USB4000 spectrometer. Light-red and green shadings show areas of overlapping between red and green rod visual pigment spectrum and solar emission, respectively (differs about two-fold). The measurement of solar spectrum was performed by the late V.I. Govardovskii.

**Table 1 ijms-22-13400-t001:** The parameters defining the light capture properties of anuran green and red rods. Here, a_max_ is T-density of visual pigment at its maximum absorption wavelength λ_max_. Factor f = (1 + L/T)/2 is a correcting coefficient for nonpolarized light illumination, where L/T is the dichroic rate of the visual pigment. S(λ_LED_) is a relative sensitivity of visual pigment (calculated from Govardovskii et al. template [38]) to a stimulating LED used in our electrophysiological setup. Data are presented as the means ± SD. The number of cells in groups are given in parentheses.

	Frog		Toad	
	Red Rod	Green Rod	Red Rod	Green Rod
a_max_, µm^−1^	0.014 ± 0.003 (*n* = 75)	0.012 ± 0.002 (*n* = 90)	0.019 ± 0.003 (*n* = 33)	0.017 ± 0.002 (*n* = 33)
f	0.62 ± 0.02 (*n* = 20)	0.64 ± 0.03 (*n* = 20)	0.64 ± 0.03 (*n* = 23)	0.66 ± 0.04 (*n* = 24)
λ_max_, nm	502	433	503	432
S(λ_LED_)	0.749	0.025	0.761	0.023

**Table 2 ijms-22-13400-t002:** The parameters extracted from analysis of green and red rods responses to brief flashes. Here, *h* is the Hill coefficient, and *I*_0.5_ is the half-saturating flash intensity. The minimum phototransduction delay was defined from the responses to bright flashes (0.5–1 R*/μm^3^), and the integration time is the integral of normalized unsaturated response over its duration time. Activation rates were defined relative to average response of frog/toad RR response (details in text). Turn-off time constants were calculated from single exponential fit of unsaturated photoresponse decay phase. *—statistically significant differences from red rods of respective species according to t-test with Welch’s correction (*p* < 0.05). Data are presented as the means ± SD. The numbers of cells in groups is given in parentheses.

	Frog		Toad	
	Red Rod	Green Rod	Red Rod	Green Rod
Dark current, pA	19.0 ± 6.5 (*n* = 12)	7.6 ± 2.9 * (*n* = 12)	15.2 ± 4.2 (*n* = 8)	12.3 ± 3.9 (*n* = 9)
*I*_0.5_, R*/μm^3^/flash	0.021 ± 0.008 (*n* = 12)	0.011 ± 0.003 * (*n* = 12)	0.013 ± 0.006 (*n* = 8)	0.009 ± 0.004 (*n* = 9)
*h*	1.27 ± 0.17 (*n* = 12)	1.24 ± 0.14 (*n* = 12)	1.11 ± 0.13 (*n* = 8)	1.13 ± 0.19 (*n* = 9)
Minimum delay, ms	12.7 ± 1.6 (*n* = 7)	13.2 ± 0.8 (*n* = 5)	12.7 ± 2.2 (*n* = 6)	14.6 ± 2.7 (*n* = 5)
Relative activation rate	1.02 ± 0.39 (*n* = 8)	1.02 ± 0.49 (*n* = 10)	0.98 ± 0.46 (*n* = 6)	1.37 ± 0.36 (*n* = 8)
Time to peak, s	0.55 ± 0.08 (*n* = 8)	0.97 ± 0.22 * (*n* = 11)	0.84 ± 0.33 (*n* = 6)	1.39 ± 0.27 * (*n* = 8)
Integration time, s	1.17 ± 0.41 (*n* = 8)	1.68 ± 0.29 * (*n* = 11)	1.55 ± 0.55 (*n* = 6)	2.88 ± 0.72 * (*n* = 9)
Turn-off time constant, s	0.89 ± 0.32 (*n* = 8)	1.27 ± 0.38 * (*n* = 10)	0.94 ± 0.4 (*n* = 6)	2.28 ± 0.62 * (*n* = 9)

**Table 3 ijms-22-13400-t003:** Anuran green and red rods discrete dark noise rate and visual pigment stability. SPR–single photon response. *—statistically significant differences from red rods of respective species according to t-test with Welch’s correction (*p* < 0.05). Data are presented as the means ± SD. The numbers of cells in groups is given in parentheses.

	Frog		Toad	
	Red Rod (*n* = 8)	Green Rod (*n* = 12)	Red Rod (*n* = 11)	Green Rod (*n* = 8)
Corrected SPR amplitude, pA (% from dark current)	0.44 ± 0.25 (2.3 ± 0.8%)	0.26 ± 0.1 (4.1 ± 1.5% *)	0.77 ± 0.61 (4.6 ± 3.1%)	0.49 ± 0.22 (5.7 ± 3.6%)
Discrete events frequency per cell, s^−1^	0.031 ± 0.005	0.024 ± 0.009 *	0.028 ± 0.005	0.021 ± 0.008
Dark events rate per unit OS volume * 10^−6^, μm^−3^ * s^−1^	13.2 ± 2.2	18.8 ± 6.9 *	12.6 ± 2.4	13.6 ± 5.2
Average spontaneous isomerization rate constant * 10^−11^, s^−1^	0.71 ± 0.12	0.89 ± 0.33	0.67 ± 0.13	0.65 ± 0.25

## Data Availability

The data reported in this study are available upon request from the corresponding author and are not available to the public because of their size.

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
