# Peer review of "Phototransduction in Anuran Green Rods: Origins of Extra-Sensitivity"

_ijms, 2021, doi:10.3390/ijms222413400_

Round 1

Reviewer 1 Report

In this paper, the authors compare the suction pipette recordings of the green rods of common frog (Rana temporaria) and common toad (Bufo bufo), by examining their light sensitivity, response kinetics and parameters of discrete and continuous dark noise. Although this research uses quite dated techniques and data analysis, it could be worth to publish providing the authors answer to the following inquiries and make the asked corrections.

Major:

1) At line 575 the authors state that “All records were done on single isolated rods with outer or inner segment drawn into the pipette”. When the outer segment is drawn into the suction pipette then stimulating light could be scattered and/or absorbed by the pipette glass, but this does not happen when the inner segment in drawn into the pipette. Moreover, the current amplitude is just a fraction of the real one (usually about 70%) because the seal between the pipette and the cell is not perfect, and this seal is usually larger in the “outer segment” recordings in respect to the "inner segment" ones (i. e. the fraction of the “real” current is larger in the former recordings in respect to the latter ones).  How the authors evaluate the stimulus intensities and the current amplitudes (whose statistics is reported in table 2) in the two recording configurations?

2) in light of the inquiry 1), and on the fact that, looking at the statistics, the authors have a large variability in their recordings (for instance, the activation rate of red and green rods from the steepness of the fractional photoresponse rising phase of fig. 4 varies of a factor 3-4), how much the responses of fig. 2 are representative? Are they the “best” cells, as reported at line 267? This criticisms applies also to Fig. 3A, 3B, 4A, 5A, 5B, 6A, 6B, and A1: are all these recordings caming from the same green and red rods?

3) At line 66-67 the authors state that “Toad’s GRs also resembles typical RRs by generating discrete dark noise – spontaneous current waves in complete darkness which are identical in shape to the responses to single photon”. Besides that I would put a “-“ after “darkness”, how the authors reconcile this statement with  the one at line 488 (and following): “…our group previously reported that the amplitude of discrete dark events in anuran rods varies much greater than that of SPRs. If so, we shall assume that variation of dark events size may has a different origin and follows a different laws, than SPR does.” I do not think that these variable “discrete dark events” are generated by the spontaneous activation of transduction cascade, but it is instead a low frequency noise generated by the calcium negative feedback regulating the response amplitude: any delay in this feedback loop exerted by the internal calcium buffer(s) enhances this low frequency noise (see, for instance: DOI: 10.1039/b303871h). Therefore, I think that the study on toad green rods by Luo et al. [29], that gave a value of the spontaneous isomerization frequency about 150 times lower than the one reported by the authors, is correct.

Minor:

Line 14: “of the two anuran species”: which species? Only at line 91-92 it is specified that they are common frog (Rana temporaria) and common toad (Bufo bufo)

Line 323: in “R*/μm−3 per flash” the “-3” must be superscript

Line 492-493: “... then the size of dark events varies within relatively wide scopes, instead of being almost fixed.” I do not understand “relatively wide scopes” referred to the size of something

Reviewer 2 Report

The manuscript by Astakhova and colleagues presents a series of carefully designed and performed experiments seeking to characterize the properties of amphibian green rods that express a blue cone visual pigment but have rod-like morphology. They achieve that by comparing the responses of green rods with these of red rods in two amphibian species, toad and frog. Consistent with previous studies, they find that green rods have high, rod-like sensitivity and are indeed slightly more sensitive than red rods both in darkness and in background light. They also demonstrate that this higher sensitivity is the result of slower response inactivation of green vs. red rods, whereas activation rates appear comparable for the two photoreceptor types. Finally, using dark noise measurements, the authors demonstrate that red and green rod visual pigments have comparable rates of thermal activation.

Overall, this is a very well designed and executed study. The experiments are carefully designed and rigorously described and performed, with logical and convincingly presented results. The Introduction and Discussion and scholarly and thought-provoking, with balanced overall analysis of the implications of the findings in the context of previous results (but see below for minor comments).  The paper is a pleasure to read and represents a valuable and scholarly contribution to the field. While I do not have any major concerns, addressing a few minor points will improve this already excellent study.

Minor points:

  • When determining the spontaneous isomerization rates for green and red rods in frogs and toads, the authors use pigment concentrations calculated previously by Harosi. Considering that the published values for the two rod types differ by about 13%, this value could possibly affect the overall conclusions, especially since the dark events rate per unit OS (Table 3) seem slightly higher in green rods for both species. Could the authors use their own MSP data to determine the pigment concentrations of the cells used in this study rather than relying on published data? This could provide more accurate estimates, possibly affecting their conclusion about the relative rates of thermal isomerization of red and green visual pigments.
  • Although the authors mention the study of salamander green rods by Ma and colleagues (ref 14), it is only in passing. However, this study bears a lot of similarity to the current work, both in methodology and in insights about the function of green rods. The authors should discuss more explicitly and extensively the parallels and differences between their findings and these of Ma and colleagues from salamander green rods.
  • Similarly, the rate of thermal activation of green rod pigment has been characterized previously by Rieke and Baylor. The authors mention that study to introduce the concepts of discrete and continuous noise (ref 45) but should discuss the parallels between their findings in green rods with these of Rieke and Baylor from salamander blue cones that share the same visual pigment with green rods.
  • The authors imply that green rods might be involved in facilitating color discrimination in scotopic conditions by working together with red rods. Such color discrimination requires not only photoreceptors with different spectral sensitivities, but also the downstream retinal circuitry required to compare the signals from the two rod types. It will be helpful to comment in the Discussion what, if anything, is known about the retinal wiring of green rods and how their signals are processed in the retina compared to the signals from red rods.

Reviewer 3 Report

Dear Authors,

Manuscript ID: ijms-1468189, "Phototransduction in anuran green rods: origins of extra-sensitivity" is interesting to peers. The data is straightforward that is based on spectral properties of visual pigments and flashes in response to rods. The claim that "GRs are efficient and more specialized single-photon detectors than RR" sounds pretty good and is convincing. However, my concern is that the transmutation from ancestral cones, isomers of cone-rod, and the mesopic effect may cause them to decline as complete nocturnal photoreceptors. Also, I got lost in the draft while attempting to conclude their significant findings due to a lot of exaggerated text. Some of my suggestions are as follows:

  1. Excessive exaggeration in the introduction. Would you please shorten it to be concise and data-specific?
  2. Results 2.1. Keep it brief and avoid delving into the "effective light intensity" formula mentioned in [34]. Avoid unnecessary explanations and concentrate solely on the results. Rather than mentioning multiple spectrums, please specify the exact number used to average the spectrum recording. Similarly, other results must be concrete to be transparent and clear.
  3. Please keep the discussion brief and to the point.
  4. Please include the study's limitations.
  5. Include statistical analysis in supplementary data for all plots.

Best Wishes and Thank you

Reviewer 4 Report

The article by Astakhova et al. describes the responses of red and green rods in frog and toad retinas. Whereas, the green rods have been described in toads more extensively before, this is the first time the physiological properties of the two rods have been compared in two anuran species. 

I have some minor comments on the article-

  1. Lines 102-105: the properties of green rods in frogs were first described by JK Bowmaker in 1976-1977. Please modify the statement and include the citations for these papers.
  2. Were there any observable differences in the spectral properties of red or green rods that were temperature sensitive? Kristian Donner’s group have previously found differences in sensitivity of red rods in toads with changing temperatures.

Round 2

Reviewer 1 Report

The authors have satisfactorily answered to all my comments and suggestions and, as far as I’m concerned, the paper is worth to be published in the International Journal of Molecular Sciences as it is. 

Reviewer 3 Report

Dear Authors,

Thank you for taking the time to address all of my comments and for making the manuscript as clear and straightforward as possible. Now I can recommend it for publication.

Best Wishes